∂ | **Open Peer Review** | Virology | Research Article

# High pathogenicity avian influenza A (H5N1) clade 2.3.4.4b virus infection in a captive Tibetan black bear (*Ursus thibetanus*): investigations based on paraffin-embedded tissues, France, 2022

Pierre Bessière,[1] Nicolas Gaide,[1] Guillaume Croville,[1] Manuela Crispo,[1] Maxime Fusade-Boyer,[1] Yanad Abou Monsef,[1] Malorie Dirat,[1] Marielle Beltrame,[2] Philippine Dendauw,[2] Karin Lemberger,[3] Jean-Luc Guérin,[1] Guillaume Le Loc'h[1]

**ABSTRACT**   High pathogenicity avian influenza viruses (HPAIVs) H5Nx of clade 2.3.4.4b have been circulating increasingly in both wild and domestic birds in recent years. In turn, this has led to an increase in the number of spillover events affecting mammals. In November 2022, an HPAIV H5N1 caused an outbreak in a zoological park in the south of France, resulting in the death of a Tibetan black bear (*Ursus thibetanus*) and several captive and wild bird species. We detected the virus in various tissues of the bear and a wild black-headed gull (*Chroicocephalus ridibundus*) found dead in its enclosure using histopathology, two different *in situ* detection techniques, and next-generation sequencing, all performed on formalin-fixed paraffin-embedded tissues. Phylogenetic analysis performed on the hemagglutinin gene segment showed that bear and gull strains shared 99.998% genetic identity, making the bird strain the closest related strain. We detected the PB2 E627K mutation in minute quantities in the gull, whereas it predominated in the bear, which suggests that this mammalian adaptation marker was selected during the bear infection. Our results provide the first molecular and histopathological characterization of an H5N1 virus infection in this bear species.

**IMPORTANCE**   Avian influenza viruses are able to cross the species barrier between birds and mammals because of their high genetic diversity and mutation rate. Using formalin-fixed paraffin-embedded tissues, we were able to investigate a Tibetan black bear's infection by a high pathogenicity H5N1 avian influenza virus at the molecular, phylogenetic, and histological levels. Our results highlight the importance of virological surveillance programs in mammals and the importance of raising awareness among veterinarians and zookeepers of the clinical presentations associated with H5Nx virus infection in mammals.

**KEYWORDS**    influenza, zoonotic infections, epidemiology

The genetic and antigenic diversity of influenza viruses is considerable: 16 and nine hemagglutinin (HA) and neuraminidase subtypes, respectively, circulate in wild waterfowl, which were considered the reservoir of influenza viruses (1). Following the acquisition of a mutation in the sequence encoding the HA cleavage site, viruses belonging to the H5Nx and H7Nx subtypes are capable of acquiring a high pathogenicity phenotype in poultry. The tropism of low pathogenicity avian influenza viruses is mainly restricted to the digestive and respiratory tracts, while high pathogenicity avian influenza viruses (HPAIVs) can replicate systemically (2).

For a long time, HPAIVs circulating in domestic birds were considered unlikely to return to the wild compartment (3). However, in recent years, this paradigm has been shaken: HPAI H5Nx viruses, which initially appeared in the domestic compartment, have

Address correspondence to Pierre Bessière, pierre.bessiere@envt.fr.

The authors declare no conflict of interest.

succeeded in becoming endemic in wild birds (4). More importantly, these viruses have managed to cross the species barrier between mammals and birds on several occasions.

H5Nx HPAIVs' circulation, particularly those of clade 2.3.4.4b, has significantly increased in recent years: viruses of this clade are spreading in wild bird populations more rapidly than ever since their emergence in 1996 (5). Thus, the occasions on which mammals have been exposed to the virus have become more frequent in turn. Infection generally occurs following the exposure to contaminated feces or water or animal carcasses (6). Sporadic infections of wild mammals have been described in the past but have never been as frequent as in recent years. Numerous domestic and wild animals have been contaminated, including various species of seals, sea lions, foxes, cats, raccoons, skunks, and bears (7–12).

These sporadic infections are not always dead ends (13), and every time an avian influenza virus manages to infect a mammal, there is a risk that adaptive mutations will appear (1). When enough genetic changes accumulate, the result can be the emergence of a virus more efficiently transmitted between mammals. Ultimately, a novel influenza virus able to sustain transmission between humans could cause a pandemic (14).

In November 2022, a Tibetan black bear (*Ursus thibetanus*) from the Sigean Zoo in France was reported dead. Although infection with an avian influenza virus was not initially suspected, post-mortem analyses revealed the presence of clade 2.3.4.4b HPAIV H5N1 in various organs and blood. Over the following fortnight, several influenza-positive birds were also found dead near the bear enclosure, suggesting a bird-to-bear transmission. Using formalin-fixed paraffin-embedded (FFPE) tissue, we conducted molecular and histopathological investigations to characterize this outbreak.

## RESULTS

### Outbreak detection

In early November 2022, a 12-year-old male Tibetan black bear (*U. thibetanus*) was found dead in its enclosure at the Sigean Zoo, in France. A few days before, zookeepers had noticed a slight decline in general condition, while the day prior to its death, the bear presented marked dyspnea, hyperthermia (rectal temperature: 39°C), lateral decubitus, and diarrhea. Blood analyses carried out by the zoo's veterinarians the day before the bear's death showed severe leukopenia and hypercreatininemia, consistent with acute renal failure (File S1). Over the following days, several zoo and wild birds (two pelicans, one jackdaw, and one gull) died and tested positive by reverse-transcription quantitative PCR for clade 2.3.4.4b HPAIV H5N1, one of the carcasses being found in the bear's enclosure: a black-headed gull (*Chroicocephalus ridibundus*), which the zoo veterinarians also necropsied. In the days following the bear's death, other bears of the same species housed in the same enclosure displayed mild to moderate clinical respiratory signs but could not be sampled to be analyzed as part of this study. Interestingly, none of the bears in the nursery (i.e., not in this enclosure and without access to the outdoors), at this time, developed clinical signs. A summary of animals affected by this epizootic is available in Table S1.

The possibility that the bear might have been infected by an H5N1 virus of clade 2.3.4.4.b was subsequently suspected and confirmed by molecular analysis of biological samples sent to the French reference laboratory for high pathogenicity avian influenza, which deposited the viral genome sequence on GISAID (isolate ID EPI_ISL_17233426). After initial routine screening at a diagnostic histopathology laboratory (Vet Diagnostics, France), FFPE tissues from the bear and gull were then sent to the National Veterinary College of Toulouse (ENVT) Laboratory for further investigation.

## Pathological examination of the infected bear and black-headed gull

The bear necropsy revealed hemorrhagic lesions (petechiae and suffusion) on the epicardium and liver, severe congestion of lungs, kidneys and intestines, and finally, a necrotic tracheal mucosa. Importantly, no bird remains were found in the digestive tract. Histopathological examination of the bear revealed acute, marked multifocal to coalescing fibrino-necrotizing lymphadenitis and splenitis, associated with vasculitis and hemorrhages (Fig. 1); marked pulmonary edema and congestion; moderate suppurative tracheitis with concurrent submucosal vascular thrombosis; and severe, multifocal necro-suppurative hepatitis. Marked acute hemorrhages, involving the subepicardium and renal interstitium, were also present, while the gastrointestinal tract appeared unremarkable (Fig. S1).

The gull necropsy revealed a marked pulmonary and splenic congestion, while the histopathological assessment showed acute necrotic-inflammatory changes in the majority of the organs examined. Mild to marked encephalitis, pancreatitis, splenitis, hepatitis, nephritis, and thyroiditis lesions exhibited variable amounts of viral antigen and RNA highlighted by immunohistochemistry (IHC) and RNAscope *in situ* hybridization (RNAscope ISH), respectively (Fig. S2 and S3).

Viral antigen and RNA detections of the bear are detailed in Table S2. Briefly, viral antigen and RNA were frequently detected in a visceral lymph node, multifocally, within perivascular tissue (leukocytes) adjacent to foci of necrotizing vasculitis (Fig. 1). In the lung, viral antigen was sparsely detected within bronchoalveolar luminal debris and perivascular and interlobular interstitium (mesenchymal cell) admixed with non-specific background. Viral RNA was similarly detected in terms of distribution, although the signal appeared more widespread within the pulmonary parenchyma, also involving alveolar epithelial cells, subpleural mesenchymal cells, and mesothelial cells (Fig. 1). In the kidney, viral antigen was rarely observed within a few glomeruli (mesangial and endothelial cells), while viral RNA detection was negative (Fig. S1). Additionally, RNAscope ISH revealed sparse viral RNA within myocardial endomysium and perimysium (mesenchymal cells), gastrointestinal and tracheal mucosa and submucosa, and splenic red pulp. Other findings included focal atherosclerosis and mural mineralization involving the coronary arteries of cardiac sections, with no evidence of related myocardial ischemic changes.

Viral antigen and RNA detection of the gull are detailed in Table S3. Viral antigen detection appeared widespread in the optic lobe, cerebellum (neurons, glial cells, Purkinje cells, and neuropil), and pancreas (acinar cells), frequent and multifocal in the heart (cardiomyocytes), pulmonary capillary bed, and kidney (tubular nephrocytes). The spleen, intestine, liver, and trachea were negative. Viral RNA detection exhibited a similar pattern, in terms of distribution, in the optic lobe, cerebellum, and kidney with additional positive staining in a renal nervous ganglion.

## Phylogenetic and genetic analyses

We performed next-generation sequencing on several bear samples (lymph node, lung, and liver) and one gull sample (brain), using an Element AVITI sequencer (Element Biosciences, San Diego, CA, USA) and a 2 × 150-bp paired-end protocol. We found H5N1 virus reads in all samples, in sufficient numbers to reconstitute whole-genome consensus sequences. The H5N1 virus was the only pathogen detected by next-generation sequencing. Details of the viral (>100 reads) and bacterial (>1,000 reads) species identified by metagenomics can be found in File S2.

Phylogenetic analysis performed on HA gene segment showed that viruses found on both animals belonged to clade 2.3.4.4b. The HA sequence of the bear-derived virus shared 99.998% genetic identity with the HA of the strain detected in the gull, making the bird strain the closest related strain (Fig. 2). This finding was further confirmed by the phylogenetic analysis performed on the other viral segments, which showed that

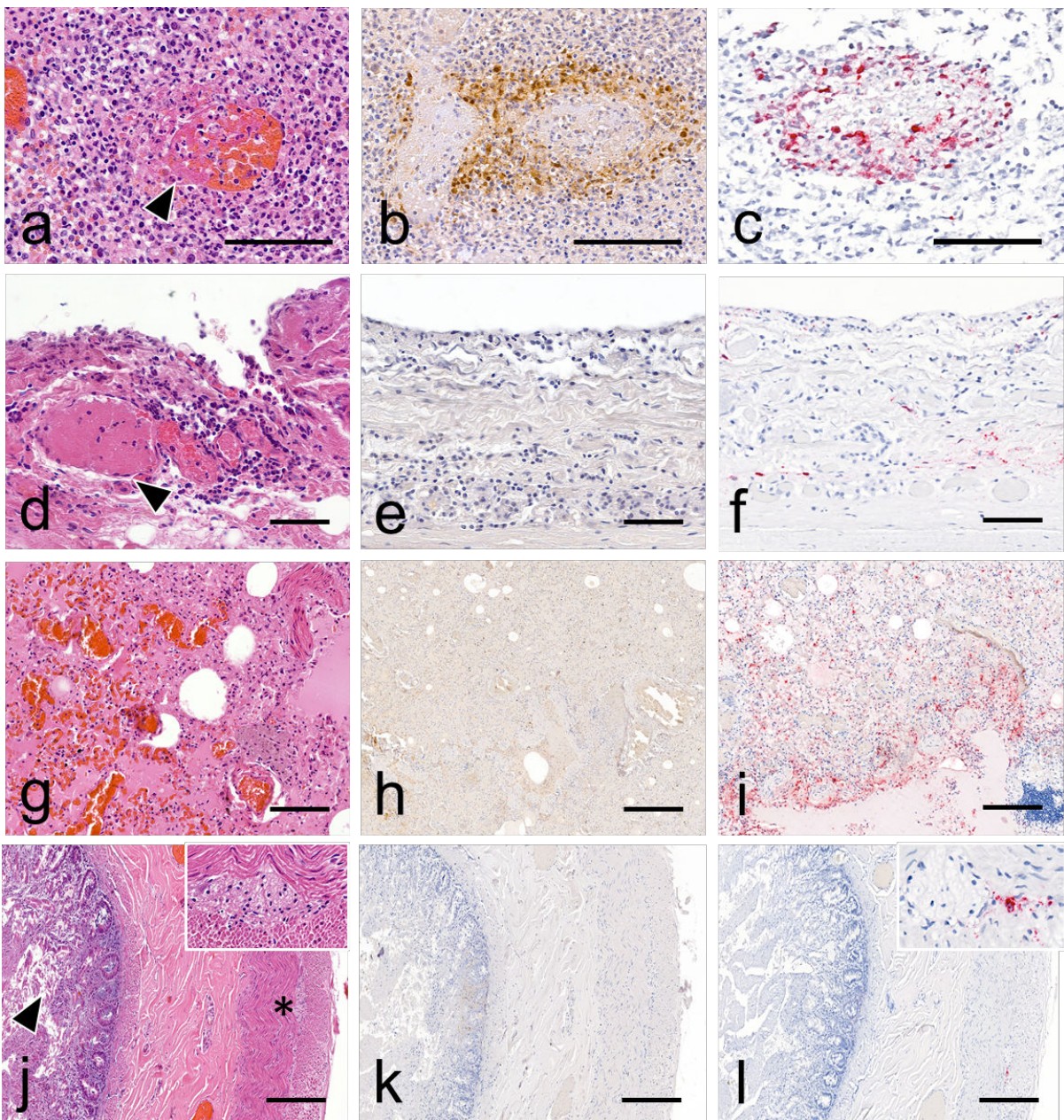

**FIG 1** Histopathology, viral antigen, and RNA detection in tissues obtained from infected bear. (a) Visceral lymph node: necrotizing vasculitis (arrowhead) with thrombosis and hemorrhages. H&E stain. (b) Visceral lymph node: viral antigen is observed associated with vasculitis and extending to the surrounding lympho-nodal parenchyma. Anti-nucleoprotein influenza A IHC (anti-NP IHC). (c) Visceral lymph node: viral RNA is intralesionally detected within areas of vasculitis. M gene RNAscope ISH. (d) Trachea: thrombosis (arrowhead) and perivascular leukocytic infiltration are observed within the mucosa and submucosa. The overlying epithelium is sloughed (arrowhead) (H&E stain). (e) Trachea: no viral antigen detection is observed (anti-NP IHC). (f) Trachea: positive viral RNA detection is observed in the interstitium of mucosa and submucosa (RNAscope ISH). (g) Lung: diffuse congestion and edema (H&E stain). (h) Lung: IHC shows moderate non-specific background staining with no significant detection of viral antigen at low magnification (anti-NP IHC). (i) Lung: viral RNA is widely distributed within the lobular and interlobular interstitium (RNAscope ISH). Intestine: autolytic changes are present in the mucosa, including cell sloughing (arrowhead). The submucosa, tunica muscularis, and serosa appear within normal limits. The myenteric plexus (insert and asterisk) is readily identifiable and also normal (H&E stain). (j) Intestine: no viral antigen detection is observed (anti-NP IHC). (l) Intestine: viral RNA is focally associated with the myenteric plexus nerve trunk and connective tissue (insert) (RNAscope ISH). Scale bars: 50 (d–f), 100 (a–c and g–i), and 200 µm (j–l).

both strains were closely related to strains circulating in Belgium at that time (Fig. S4). The whole-genome analysis revealed that both strains belonged to the AB genotype

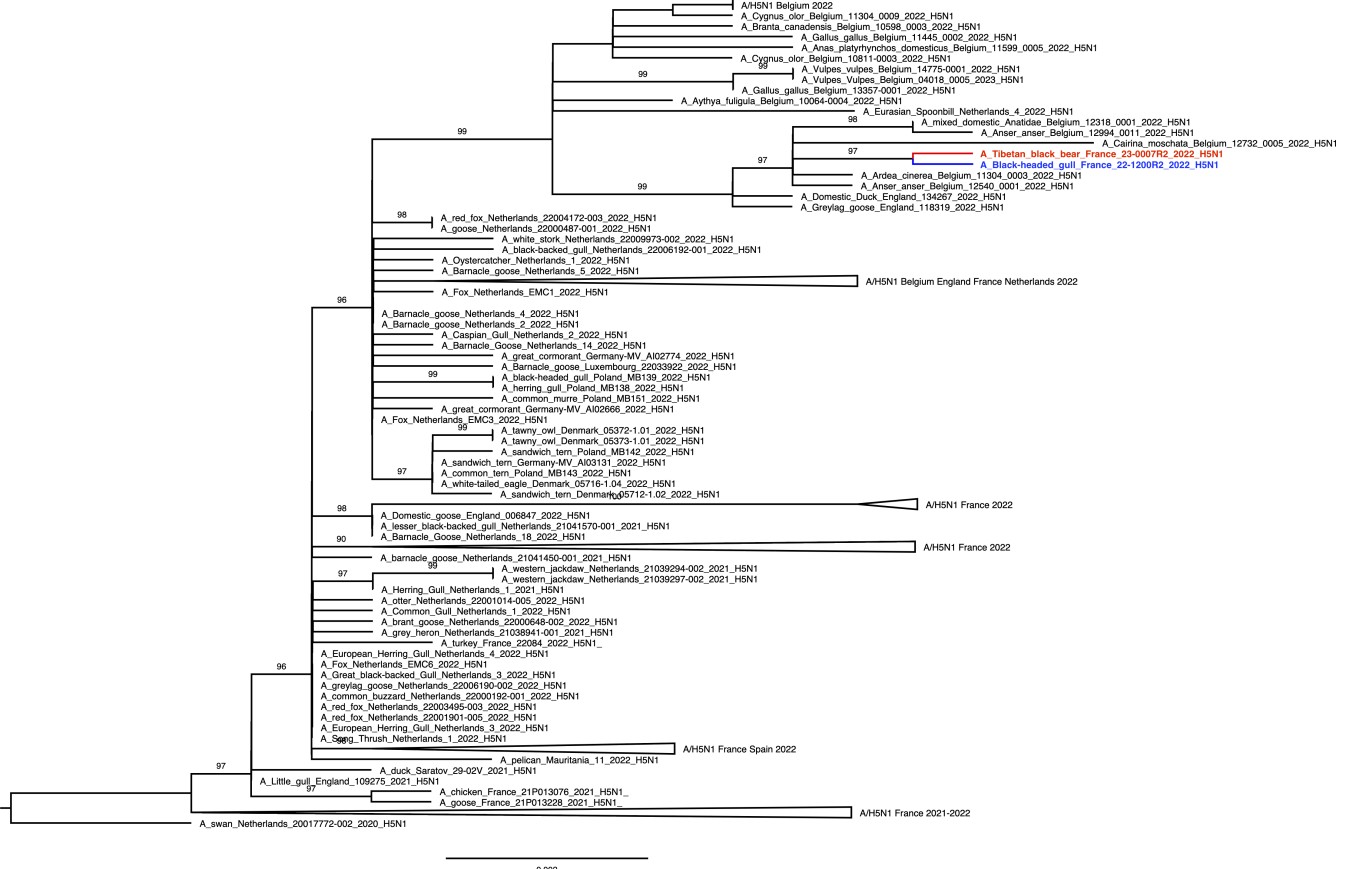

**FIG 2** HA maximum likelihood phylogenetic tree. Bear- and gull-derived sequences are labeled in red and blue, respectively. Bootstrap support values > 90 are shown at branches. Scale bar: number of nucleotide substitution per site.

(H5N1-A/duck/Saratov/29–02/2021-like), the main circulating genotype at that time in Europe (6).

Viral sequence analysis revealed the presence of several mammalian adaptation markers, listed in Table 1, in both bear and gull viruses, most of them also found in the genomes of other phylogenetically related avian viruses. The PB2 E627K mutation was only found in the bear virus according to the consensus sequences, but variant calling analysis revealed that a tiny proportion of gull viruses also had this mutation (File S3). More specifically, in the gull, over 99% of reads coded for an E and <1% coded for a K, while in the bear, 37.3% coded for an E and 62.7% coded for a K at position 627, both in

**TABLE 1** Mammalian adaptation markers found on the gull- and bear-derived viral sequences[a]

| Protein | Amino-acid position | Gull | Bear | Reference |
|---------|---------------------|------|------|-----------|
| PB2 | 627 | E | K | Hatta et al. (15) |
| PB1-F2 | 66 | S | S | Varga et al. (16) |
| HA | 149 | A | A | Yang et al. (17) |
| | 170 | N | N | Wang et al. (18) |
| | 172 | A | A | |
| M1 | 30 | D | D | Fan et al. (19) |
| | 215 | A | A | |
| NS1 | 42 | S | S | Jiao et al. (20) |
| | 92 | D | D | Seo et al. (21) |
| | 103 | F | F | Kuo and Krug |
| | 106 | M | M | (22) |

[a]HA sequence numbering was performed including the peptide signal.

the liver and lung samples. Although the sequencing depth at this position was around 1,400 reads, no sequence encoding an E at position 627 was found in the lymph node sample.

## DISCUSSION

We revealed the presence of clade 2.3.4.4b H5N1 virus in various tissues of a Tibetan black bear and a black-headed gull, using histopathology, two different *in situ* detection techniques, and next-generation sequencing. Sequence analysis revealed that the viruses found in the bear and gull were phylogenetically very close and that variants with the PB2 E627K mutation had become predominant in the bear. Our findings do not allow us to determine with any certainty how the bear became infected, especially as the gull was found dead several days after the bear. The most plausible hypothesis is that the virus circulated undetected for several days in the zoo's wild and captive avifauna and that the bear was either in direct contact with an infected bird (dead or alive) or in contact with water or food contaminated by avian droppings. The fact that the other bears in the enclosure successively developed clinical signs suggests inter-bear transmission, but since they were not sampled for PCR testing, this is speculation. Interestingly, the PB2 E627K mutation was present in very small amounts in the gull sample, whereas it was predominant in the bear lung sample. We, therefore, believe that it did not arise in the bear through *de novo* mutations, but that the bear's infection was seeded by an inoculum containing PB2 627E variants and a few PB2 627K variants, which were then selected because of their selective advantage (15). Interestingly, the bear lymph node sample contained no PB2 627E variants. This suggests that the PB2 627K variant was selected in the respiratory tract, from which it then spread systemically.

When an animal is necropsied outside a research facility, improper preservation of samples is a frequent issue, especially if the carcasses cannot be refrigerated and if the analysis cannot be performed quickly. One of the advantages of FFPE specimens is that they can be stored at room temperature for years, if not decades, and are shipped easily (23). Although this process degrades the nucleic acids to some extent, methods for extracting DNA and RNA of sufficient quality are now available (24). This is how we managed to use FFPE tissues to reconstruct an H5N1 avian influenza virus outbreak in a zoological park through molecular investigations, despite the absence of fresh tissue samples.

One limitation of our study is that the bear's brain was not examined and sampled. Collecting this organ would have been technically challenging, and since the zoo's veterinary service did not suspect an infection with an avian influenza virus at the time of the necropsy, the cranium was not opened. However, the involvement of the central nervous system has been frequently identified in both birds and mammals infected with an H5Nx virus of clade 2.3.4.4b, with neurological disorders sometimes being the only clinical signs observed (25). For this reason, highlighting the presence of the virus and concurrent histopathological lesions in the brain of our bear would have been of particular interest. However, the bear was not affected by neurological disorders, contrary to what was reported in Canada in black bears (*Ursus americanus*) (12).

In zoological parks, many animal species may reside in close proximity to one another, and avoiding contact with wild birds is difficult, if not impossible, when animals are not kept in cages, but in open spaces, as it is the case with bears in the Sigean Zoo. Zookeepers visited the bears' enclosure on a daily basis, but both the enclosure's size and the density of the vegetation made most biosecurity measures virtually impractical. On a broader level, zookeepers should receive proper training in biosecurity and made aware of the threats posed by avian influenza viruses toward mammals. This aspect is important, not only for protecting animals, particularly in the case of endangered species, but also for preventing epizootic flare-ups. As the RNA polymerase of influenza viruses is prone to errors during viral genome replication, better-adapted variants may appear when a mammal is infected by an avian influenza virus (26). When such a virus spreads from mammal to mammal, the risk of a more transmissible variant being

selected increases considerably, and chains of transmission must be avoided as much as possible (27). By raising awareness among veterinarians and zookeepers of the clinical presentations associated with H5Nx virus infection in mammals, the number of undetected epizootics should be reduced, with zoological parks thus acting as sentinels.

In conclusion, biosecurity and surveillance programs are essential to deal with epizootics caused by clade 2.3.4.4b H5Nx viruses and the zoonotic spillovers that are becoming increasingly frequent. In particular, active and passive surveillance of mammals, including wild, captive, and domestic, will be invaluable in anticipating the emergence of H5Nx viruses with pandemic potential (28).

## MATERIALS AND METHODS

### Histopathology

Tissue samples were placed in 10% neutral buffered formalin. For the bear, available tissues included the trachea, lung, heart, visceral lymph node, spleen, intestine, stomach, and kidney. For the gull brain, trachea, lung, heart, spleen, pancreas (splenic lobe), thyroid gland, liver, intestine, and kidney were collected. After fixation, tissues were routinely processed in paraffin blocks, sectioned at 3 µm, stained with hematoxylin and eosin (H&E), and examined by light microscopy.

### Immunohistochemistry

To assess viral antigen tissue distribution within the bear and gull tissues, IHC was performed on FFPE tissue sections, using a monoclonal mouse anti-nucleoprotein influenza A virus antibody (Biozol BE0159, pronase 0.05% retrieval solution, 10 min at 37°C: antibody dilution 1/2,000, incubation overnight at 4°C). The immunohistochemical staining was revealed with horseradish peroxidase (HRP)-labeled polymer (EnVisio + Dual Link System HRP, K4061, Agilent) and the diaminobenzidine HRP chromogen (DAB + liquid, K3467, Agilent). Negative controls included sections incubated either without the primary antibody or with another monoclonal antibody of the same isotype (IgG2).

### RNAscope *in situ* hybridization

To determine the presence of avian influenza A virus RNA and assess its distribution within the bear tissue sections, RNAscope ISH was performed as previously described (29). Briefly, we used probes targeting M1 and M2 genes (V-InfluenzaA-H5N8-M2M1 probe), H5 HA gene (V-InfluenzaA-H5N8-HA-O1 probe) of clade 2.3.4.4b HPAIV H5, and an RNAscope 2.5 high-definition red assay, according to the manufacturer's instructions, including mild pretreatment conditions (15-min incubation with protease digestion for antigenic retrieval) and hematoxylin counterstaining. A probe targeting the dihydrodipicolinate reductase (dapB) gene from the *Bacillus subtilis* strain SMY served as negative control.

### Next generation sequencing

Three bear samples and one gull sample were selected for the metagenomics analysis on the RNA fraction: lymph node (bear), lung (bear), liver (bear), and brain (gull). One bear sample (liver) was selected for the metagenomics analysis on the DNA fraction.

The nucleic acids were extracted from the FFPE tissue sections using the Nucleospin total RNA FFPE XS (Macherey-Nagel). RNA-sequencing libraries were prepared using the NEBNext Ultra II Directional RNA Library Prep Kit (New England Biolabs), and the DNA library was prepared using the NEBNext Microbiome DNA Enrichment Kit (New England Biolabs). The sequencing run was then performed on the Element AVITI sequencer (Element Biosciences, San Diego, CA, USA) using a 2 × 150bp paired-end protocol.

## Bioinformatics analysis

The metagenomic data analysis was performed with Kraken2 (30). We set a threshold at 100 and 1,000 reads for the abundance of the viral and bacterial species, respectively. The reads were then mapped on an H5N1 reference genome (GISAID isolate ID: EPI_ISL_17233426) with minimap2 (31), and the consensus sequences were generated using iVar (32).

## Phylogenetic analysis

Consensus sequences of each viral gene segment detected in black bear were compared with the most related sequences available in GISAID (https://www.gisaid.org/). We then added H5N1 sequences corresponding to the strains that circulated in France and performed an alignment with MAFFT version 7 (https://mafft.cbrc.jp/alignment/server/index.html). After identification of the most suitable model for the analysis, maximum likelihood phylogenetic trees were generated using the IQ Tree software, version 1.6.12 (http://www.iqtree.org/), with 100,000 replicates using ultrafast bootstraps. Phylogenetic trees were then visualized by using FigTree version 1.4.2. (http://tree.bio.ed.ac.uk/software/figtree/).

## ACKNOWLEDGMENTS

The authors would like to thank HELIXIO SAS (Saint Beauzire—France) for performing the metagenomic analysis.

This study was performed in the framework of the "Chaire de Biosécurité et Santé Aviaires," hosted by the ENVT and funded by the Direction Générale de l'Alimentation, Ministère de l'Agriculture et de la Souveraineté Alimentaire, France.

P.B., N.G., G.C., M.C., J.L.G., and G.L.L. conceptualized and designed the experiments. P.B., N.G., G.C., M.C., M.F.B., M.D., K.L., M.B., and P.D. performed the experiments. P.B., N.G., M.C., Y.A.M., G.C., M.F.B., K.L., J.L.G., and G.L.L. analyzed the data. J.L.G. and G.L.L. acquired the funding. P.B., N.G., M.C., and G.C. drafted the manuscript. All authors reviewed and edited the final manuscript before submission.

## AUTHOR AFFILIATIONS

[1]IHAP, Université de Toulouse, INRAE, ENVT, Toulouse, France
[2]Réserve Africaine de Sigean, Sigean, France
[3]Vet Diagnostics, Charbonnières-les-Bains, France

## AUTHOR ORCIDs

Pierre Bessière http://orcid.org/0000-0001-5657-0027
Nicolas Gaide http://orcid.org/0000-0002-6407-5007
Guillaume Croville http://orcid.org/0000-0001-7577-2089
Maxime Fusade-Boyer http://orcid.org/0000-0002-7613-6751
Jean-Luc Guérin http://orcid.org/0000-0001-7770-4012

## AUTHOR CONTRIBUTIONS

Pierre Bessière, Conceptualization, Investigation, Methodology, Writing – original draft, Writing – review and editing | Nicolas Gaide, Conceptualization, Investigation, Methodology, Writing – original draft, Writing – review and editing | Guillaume Croville, Conceptualization, Formal analysis, Investigation, Writing – review and editing | Manuela Crispo, Conceptualization, Investigation, Methodology, Writing – original draft, Writing – review and editing | Maxime Fusade-Boyer, Formal analysis, Investigation, Writing – review and editing | Yanad Abou Monsef, Investigation, Writing – review and editing | Malorie Dirat, Investigation | Marielle Beltrame, Investigation, Writing – review and editing | Karin Lemberger, Investigation, Writing – review and editing | Jean-Luc Guérin,

Conceptualization, Funding acquisition, Project administration, Supervision, Validation, Writing – review and editing | Guillaume Le Loc'h, Conceptualization, Project administration, Supervision, Validation, Writing – review and editing.

## DATA AVAILABILITY

Segment sequences can be found on GenBank (accession numbers OR634756 to OR634763 and OR634764 to OR634771 for the bear and the gull isolates, respectively).

## ADDITIONAL FILES

The following material is available online.

### Supplemental Material

**Fig. S1 to S4 (Spectrum03736-23-s0001.pdf).** Additional illustrations of histopathological findings, viral antigen and RNA distribution, and maximum likelihood phylogenetic trees performed on viral segments other than HA.
**Tables S1 to S3 (Spectrum03736-23-s0002.docx).** Summary of animals affected by the epizootic, and subtissular localization of viral antigen/RNA in the tissues.
**Supplemental file 1 (Spectrum03736-23-s0003.pdf).** Bear blood analysis results.
**Supplemental file 2 (Spectrum03736-23-s0004.xlsx).** Metagenomic table from the bear (lymph node, lung and liver) and gull (lung) samples.
**Supplemental file 3 (Spectrum03736-23-s0005.xlsx).** Variant calling results from bear and gull samples.

### Open Peer Review

**PEER REVIEW HISTORY (review-history.pdf).** An accounting of the reviewer comments and feedback.

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
