## [Reviewer comments · Microbiology Spectrum]

Microbiology Spectrum

High pathogenicity avian influenza A (H5N1) clade 2.3.4.4b virus infection in a captive Tibetan black bear (*Ursus thibetanus*): investigations based on paraffin-embedded tissues, France, 2022

Pierre Bessière, Nicolas Gaide, Croville Guillaume, Manuela Crispo, Fusade-Boyer Maxime, Yanad Abou Monsef, Malorie Dirat, Marielle Beltrame, Philippine Dendauw, Karin Lemberger, Jean-Luc Guérin, and Guillaume Le Loc'h

Corresponding Author(s): Pierre Bessière, IHAP, Université de Toulouse, INRAE, ENVT, Toulouse, France

Review Timeline:

Submission Date:	November 14, 2023
Editorial Decision:	December 12, 2023
Revision Received:	January 9, 2024
Accepted:	January 9, 2024

Editor: Robert de Vries

Reviewer(s): Disclosure of reviewer identity is with reference to reviewer comments included in decision letter(s). The following individuals involved in review of your submission have agreed to reveal their identity: Takahiro Hiono (Reviewer #1)

Transaction Report:

DOI: <https://doi.org/10.1128/spectrum.03736-23>

Re: Spectrum03736-23 (High pathogenicity avian influenza A (H5N1) clade 2.3.4.4b virus infection in a captive Tibetan black bear (*Ursus thibetanus*): investigations based on paraffin-embedded tissues, France, 2022)

Dear Dr. Pierre Bessi re:

Thank you for the privilege of reviewing your work. Below you will find my comments, instructions from the Spectrum editorial office, and the reviewer comments.

Revision Guidelines

Sincerely,
Robert de Vries
Editor
Microbiology Spectrum

Reviewer #1 (Comments for the Author):

Bessiere et al. reported an important case of HPAIV infection of a captive Tibetan black bear in a zoo in France. Although the specimen availability was limited for the analysis because the authors did not suspect HPAIV infection initially, the subsequent analysis was conducted as much as possible. Considering the recent situation of HPAIV infection in carnivore mammals, the reviewer believes that this study will be an important case report in the related area. Here, the reviewer pointed out several

issues that would improve the quality of the manuscript.

Major points;

The subject in this study was represented as "Ursus thibetanus". However, there are several subspecies in this species. Please specify if possible.

Also, the virus isolate was named "A/Tibetan blue bear/..." However, Tibetan blue bear are commonly known as a synonym of "Ursus arctos". Please confirm.

L80 The detailed data on the blood chemistry and hematology would be beneficial to know the general condition of the bear. Please indicate if available.

L91 Are there any other pathogens (i.e., rabies or distemper virus) the authors investigated by targeted methods to rule out the possibility? If yes, please indicate.

L184 Please mention that viral encephalitis was observed in the study by Jakobek et al. to discuss the difference between the two studies.

L192 Please provide detailed information on how the bear was kept in the zoo. For example, can wild birds easily access the bear? Is there any possibility for the bear to contact or even consume the carcass of the infected bird? What kind of biosecurity measure was applied?

Minor points;

L38-44 From the original definition, HPAIVs are highly pathogenic to chickens.

L195-196 description on the COVID-19 does not fit the context. Please delete.

Reviewer #2 (Comments for the Author):

The manuscript High pathogenicity avian influenza A (H5N1) clade 2.3.4.4b virus infection in a captive Tibetan black bear (*Ursus thibetanus*): investigations based on paraffin-embedded tissues, France, 2022, by Bessi re et al., describes a case of HPAIV infection in a captive Tibetan black bear. Most likely the bear was infected by a wild bird, as several wild and resident birds were also found dead and infected around the same time. This is the first HPAIV report in this species.

Overall comments

The information described in the manuscript has relevance and provides further evidence that mammalian species are at risk of infection and can develop lethal disease.

The manuscript could be converted to a shorter form, and convey the information more concisely. The flow of the article is interrupted by many repetitions. Relevant information about the events is incoherent in the different parts of the manuscript. No clear time-line of the events is given, and information necessary for the interpretation of the case is only presented in the discussion. Many virological and epidemiological concepts in the introduction are incorrectly formulated.

Specific comments

Line 23. PB2 E627K mutation was found in minute quantities in the gull. Do I understand correctly that you state you found a mammalian mutation in a bird? Line 142-143, says the opposite.

29-30. Controversial way to express this concept. On what basis you state that the species barrier between birds and mammals is considerable.?

46-51. The way this is formulated is incorrect and controversial. Gallinaceae and wild waterfowl, are not species. And many viruses are not species-specific, but can infect multiple taxa, and/or have zoonotic potential. There is no dogma in science, I think that the word you were thinking is paradigm.

62-63. Based on what evidence do you say every time? So far, mammalian mutations have occurred sporadically.

69. Why AI-infection was initially not suspected? I had understood from the abstract that a black-headed gull was found dead in the bear enclosure. I read further, and more information about when the bird was found is available in the Discussion. This information should be available before in the text.

82. Any gastric or intestinal content?

82-84. Description of the macroscopic examination should go in the Pathological examination paragraph.

84-85. Could you provide more the details about the birds? Number of resident birds infected? Where were they located in the zoo? This is relevant to work out the epidemiology of the outbreak.

88-90. How many bears? were the other bears sampled for serology? Or any other mammals? Large cats are also very sensitive to infection. Any information about them?

103. Would you consider providing the histology and virology results in two separate paragraphs? Then it is in symmetry with the description of your methods.

Reviewer #3 (Comments for the Author):

This MS titled 'High pathogenicity avian influenza A (H5N1) clade 2.3.4.4b virus infection in a captive Tibetan black bear (*Ursus thibetanus*): investigations based on paraffin-embedded tissues, France, 2022' authored by Bessi re et al. reporting on AIV H5N1 infection in a TBB (*Ursus thibetanus*) is remarkable and noteworthy, and therefore of particular scientific interest since there are few reports of IAV infections in bears in general, despite the fact that bears are top predating and omnivorous, and scavenging animals, likely with ample opportunity for IAV infections, especially considering the recent upsurge in widespread presence of AIV H5N1 infected bird carcasses in nature. Furthermore, in general there are relatively few scientific reports on bears being published. This raises particular questions of scientific interest, such as, are bears in general less susceptible (or maybe underdiagnosed?) to IAV infections (with H5N1 in particular), in comparison to other reported highly susceptible terrestrial and aquatic mammalian carnivores, such as foxes, martens, mink, badgers, and sea lions? And if so, what are the possible reasons for this? So finding clues to unravel the pathogenesis and to increase knowledge about possible interspecies transmission are of paramount scientific importance. With this background I have several questions listed below regarding this report, mainly to extent and clarify clinical, pathological and possibly epidemiological data and facts.

In general:

-What was the diagnosed cause of death of the Tibetan black bear (TBB)? See also further below.

-In several inflamed organs AIV H5N1 infection was established by means of IHC/ISH, but how was the AIV H5N1 infection established as the aetiology of morbidity and mortality? Moreover, were other pathogens or conditions excluded as aetiology/cause of disease and cause of death? And more specifically; what was the extent and possible causation of the diagnosed myocardial atherosclerosis and mineralisations on the bear's morbidity and mortality? Also given the fact that other TBBs showed comparable clinical signs without mortalities. Please elaborate on this.

-Was the BHGU found dead in the bear's enclosure partly eaten to indicate that it was possibly scavenged upon? Also, were there bird remains found in the bear's stomach (or other intestines, or maybe even remains like bird feathers found in the bear's feces) or were parts/tissues of the other found dead BHGUs/birds missing to indicate scavenging? Please elaborate on this.

-The BHGU was found dead in the enclosure after the bear died, but can it be fully excluded that the BHGU carcass was not already present before the bear's death? Were there for example thorough inspections of the enclosure performed daily? Or was the dead BHGU found in an obscured location that could have been easily overseen during daily inspections? Please elaborate on this.

-Were there any nervous signs observed in the TBB (and possibly in any birds) prior to death? What was the duration and severity and character (days, hours) of clinical signs prior to death? Please elaborate on this.

-What cells specifically were infected in the various organs? In the intestinal myenteric nerve plexus? Was there neurotropism? endotheliotropism? epitheliotropism? Was there IAV NP positivity seen in nuclei and/or cytoplasm of infected cells? In order to find answers or give some indications on the route of infection and/or pathogenesis? Please elaborate on this.

More specific comments and questions (some may overlap with general questions):

L19: consider for consistency with TBB (*Ursus thibetanus*), also to include Latin name of BHGU (*Chroicocephalus ridibundus*) already here in abstract?

L22: closest related one.. unclear ending, to what/which?

L33: consider using, ..have shown to be able..

L42: The high (and low) pathogenicity phenotype is defined in chickens (poultry).

L47: Gallinaceae, consider to include also generic layman names here such as 'chickens, poultry, and/or land fowl' or similar? In order to clarify to the reader that maybe not so familiar with technical taxonomical nomenclature.

L78-80: any clinical nervous signs observed? If so what kind, severity, duration? Please provide more detail.

L80: What kind of analysis, blood? Antemortem (how long before death?) or postmortem sampled?

L80: is there an explanation for the decubitus found? Was it an acute or chronic lesion? Due to abnormal (nervous) scratching/rubbing behavior? Were the other TBBs not affected by decubitus? Was it a specific lesion?

L84: what kind of gulls? BHGUs?

L87: space

L88: other TBBs, were they housed in the same enclosure? Also dead birds found with other bears? Interspecies transmission between TBBs suspected/likely/unlikely/impossible?

L82-84: Gross lesions, can you be more specific, stomach contents? Bird remains in the stomach? Or in other segments of the GI tract? Which lymph node is meant with visceral Ln? From thorax or abdomen? Which particular Ln was it? Draining from lungs/respiratory tract, upper or lower, or draining from liver or GI tract?

L105: Which cells were positive by IHC in lung interstitium/parenchyma? Endothelium or epithelium or others? Idem glomerular tuft, what cells? Idem myocardium? Myocardocytes or endothelium or other cells? Was the vasculitis associated with NP positive endothelial cells by IHC? These are important facts of information to interpret and possibly (partly) unravel the pathogenesis.

L112: Myocardial atherosclerosis and mineralisations? What extent? And location, which blood vessels/compartments, severe enough to cause disease/death? Were these diagnosed as co-morbidities of importance?

L114-117: what cells were infected or positive for NP by IHC and/or positive by ISH? Morphologically consistent with which cells? Were the inflamed organs with intralesional presences of viral protein/RNA interpreted as the aetiology of morbidity/mortality? Ok some listed in supplement but why not include this important info in results section?

L166: Indeed, the important question is whether the TBB had contact with a dead bird as possible source of infection, so like previously is there evidence from the BHGU carcass in the enclosure or in the bear GI tract/feces to support this speculation or not?

L171-173: which lymph node? So it drained from the respiratory tract?

L192: animal species

Supplementary figures 1-3: Tissues from a Tibetan black bear, not blue bear, also in legend. And official nomenclature is black-headed gull (BHGU), not seagull, also in legend.

Dear reviewers,

We thank you for your careful reading of the manuscript. Please find below a point-by-point response.
Please note that in addition to the requested corrections, we have taken the liberty of adding bootstrap
values to the phylogenetic trees.

Sincerely yours,

Pierre Bessi re (in the name of all authors)

Reviewer #1

**Major points;**

**The subject in this study was represented as "Ursus thibetanus". However, there are several**
**subspecies in this species. Please specify if possible.**

Unfortunately, the zoo staff were unable to give us the name of the subspecies.

**Also, the virus isolate was named "A/Tibetan blue bear/..." However, Tibetan blue bear are**
**commonly known as a synonym of "Ursus arctos". Please confirm.**

Thank you for spotting this mistake. We contacted GenBank and it has now been corrected ((A/Tibetan
black bear/France/23-0007R2/2022(H5N1))).

**L80 The detailed data on the blood chemistry and hematology would be beneficial to know the**
**general condition of the bear. Please indicate if available.**

These data are available on Supplementary File 1.

**L91 Are there any other pathogens (i.e., rabies or distemper virus) the authors investigated by**
**targeted methods to rule out the possibility? If yes, please indicate.**

Next-generation sequencing of the bear's organs did not reveal the presence of any pathogen other
than influenza. Please see lines 139-140.

**L184 Please mention that viral encephalitis was observed in the study by Jakobek et al. to discuss**
**the difference between the two studies.**

The requested changes were made. Please see lines 207-208.

**L192 Please provide detailed information on how the bear was kept in the zoo. For example, can**
**wild birds easily access the bear? Is there any possibility for the bear to contact or even consume the**
**carcass of the infected bird? What kind of biosecurity measure was applied?**

The requested changes were made. Please see lines 211-214.

**Minor points;**

**L38-44 From the original definition, HPAIVs are highly pathogenic to chickens.**

The manuscript was modified accordingly. Please see lines 40-41.

L195-196 description on the COVID-19 does not fit the context. Please delete.

We deleted this part.

Reviewer #2

Line 23. PB2 E627K mutation was found in minute quantities in the gull. Do I understand correctly that you state you found a mammalian mutation in a bird? Line 142-143, says the opposite.

You understood correctly. We shortened the paragraph to make it clearer.

29-30. Controversial way to express this concept. On what basis you state that the species barrier between birds and mammals is considerable.?

For simplicity's sake (text length constraints do not allow us to go into detail), we removed this sentence.

46-51. The way this is formulated is incorrect and controversial. Gallinaceae and wild waterfowl, are not species. And many viruses are not species-specific, but can infect multiple taxa, and/or have zoonotic potential. There is no dogma in science, I think that the word you were thinking is paradigm.

We apologize for these inaccuracies. The sentence has been shortened to make it clearer.

62-63. Based on what evidence do you say every time? So far, mammalian mutations have occurred sporadically.

We're not saying that every time an avian virus infects a mammal, adaptive mutations will appear. We're just saying that there's a possibility that this will happen.

69. Why AI-infection was initially not suspected? I had understood from the abstract that a black-headed gull was found dead in the bear enclosure. I read further, and more information about when the bird was found is available in the Discussion. This information should be available before in the text.

This information is given in the next paragraph, entitled "Outbreak detection". To make things clearer, we've added a table (Supplementary table 1) summarizing the chronology of events, the species affected by this outbreak, clinical signs etc. Prior to the bear's death, no cases of high pathogenicity avian influenza virus infection had been reported in the zoo, either in wild or captive avifauna in the region. For this reason, infection with an H5N1 virus was not immediately suspected in the bear.

82. Any gastric or intestinal content?

Please see lines 97-98 (*"Importantly, no bird remains were found in the digestive tract"*)

82-84. Description of the macroscopic examination should go in the Pathological examination paragraph.

The manuscript was modified accordingly.

**84-85. Could you provide more the details about the birds? Number of resident birds infected?**
**Where were they located in the zoo? This is relevant to work out the epidemiology of the outbreak.**

Please see Supplementary table 1.

**88-90. How many bears? were the other bears sampled for serology? Or any other mammals? Large**
**cats are also very sensitive to infection. Any information about them?**

Please see supplementary table 1. As indicated in line 85 of the manuscript, these bears could not be
sampled in this study. No felines were affected.

**103. Would you consider providing the histology and virology results in two separate paragraphs?**
**Then it is in symmetry with the description of your methods.**

The manuscript was modified accordingly.

**Reviewer #3**

**-What was the diagnosed cause of death of the Tibetan black bear (TBB)? See also further below.**
**-In several inflamed organs AIV H5N1 infection was established by means of IHC/ISH, but how was**
**the AIV H5N1 infection established as the aetiology of morbidity and mortality? Moreover, were**
**other pathogens or conditions excluded as aetiology/cause of disease and cause of death?**

The H5N1 virus was the only pathogen detected by next-generation sequencing, as we now state in
lines 139-140. Moreover, antigens and viral RNA were localized within or very close to lesions. It is
therefore logical to suspect that it was the cause of the animal's death.

**And more specifically; what was the extent and possible causation of the diagnosed myocardial**
**atherosclerosis and mineralisations on the bear's morbidity and mortality? Also given the fact that**
**other TBBs showed comparable clinical signs without mortalities. Please elaborate on this.**

Based on the tissues collected from the bear, atherosclerosis was identified, focally, in the coronary
arteries of cardiac sections. No thrombosis or signs of myocardial ischemia were observed in
association with intimal deposition of lipids/foamy macrophages and mural mineralization. Despite
being considered athero-resistant (Hurt-Carneio et al 2022), bears tend to have high lipid and
cholesterol levels during hibernation (Arinell et al 2012). Atherosclerosis have been reported in animals
held in captivity, including an aged grizzly bear with a cardiac schwannoma (Miller, 2008) and a polar
bear (McOrist 2002). Both reports mentioned the occurrence of seizures prior to death and the
detection of cerebral atherosclerotic lesions and secondary changes, including thrombosis, and
hemorrhages. Little is known about the impact of captive conditions on the development of
atherosclerosis in bears. Nutritional/metabolic factors related to the captivity, as well as seasonality,
might have played a role in our case.

Overall, we consider the atherosclerotic coronary lesions identified in this case an incidental finding,
that didn't play a role in the death of the animal.

References:

Hurt-Camejo, E., & Pedrelli, M. (2022). Why are brown bears protected against atherosclerosis even
though their plasma cholesterol levels are twice that of humans?. *Clínica e Investigación en*
*Arteriosclerosis (English Edition)*, 34(6), 322-325.

Miller AD, McDonough S. Interthalamic hematoma secondary to cerebrovascular atherosclerosis in an
aged grizzly bear (*Ursus arctos horribilis*) with primary cardiac schwannoma. *J Zoo Wildl Med.* 2008;
39(4): 659–662.

Arinell, K., Sahdo, B., Evans, A. L., Arnemo, J. M., Baandrup, U., & Fröbert, O. (2012). Brown bears (*Ursus*
*arctos*) seem resistant to atherosclerosis despite highly elevated plasma lipids during hibernation and
active state. *Clinical and Translational Science*, 5(3), 269-272.

McOrist, S., Tseng, F., Jakowski, R., Keating, J., & Pearson, C. (2002). Cerebral arteriosclerosis in an aged
captive polar bear (*Ursus maritimus*). *Journal of Zoo and Wildlife Medicine*, 33(4), 381-385.

**-Was the BHGU found dead in the bear's enclosure partly eaten to indicate that it was possibly**
**scavenged upon? Also, were there bird remains found in the bear's stomach (or other intestines, or**
**maybe even remains like bird feathers found in the bear's feces) or were parts/tissues of the other**
**found dead BHGUs/birds missing to indicate scavenging? Please elaborate on this.**

This gull was found dead two weeks after the bear's death. We do not believe that this particular
animal was responsible for infecting the bear, but rather that H5N1 virus was circulating in the zoo's
avifauna (wild and captive) at the time, and that another bird was maybe involved in the transmission.
No bird remains were found in the bear's digestive tract. Also see below (lines 174-177 of this letter).

**-The BHGU was found dead in the enclosure after the bear died, but can it be fully excluded that the**
**BHGU carcass was not already present before the bear's death? Were there for example thorough**
**inspections of the enclosure performed daily? Or was the dead BHGU found in an obscured location**
**that could have been easily overseen during daily inspections? Please elaborate on this.**

Bears at the Sigean zoological park live in a large, open-air enclosure with a lot of vegetation. Contact
with wild birds was entirely possible, and biosecurity measures could not be implemented to prevent
this. Similarly, it is possible that bird corpses may have escaped staff attention, due to the dense
vegetation. Please see lines 211-214.

**-Were there any nervous signs observed in the TBB (and possibly in any birds) prior to death? What**
**was the duration and severity and character (days, hours) of clinical signs prior to death? Please**
**elaborate on this.**

No neurological disorders were observed in the bear. No clinical signs were observed in the gull (this
was a wild animal, not a zoo captive). For a detailed description of the bear's clinical signs, see lines
76-78 and Supplementary table 1.

**-What cells specifically were infected in the various organs? In the intestinal myenteric nerve plexus?**
**Was there neurotropism? endotheliotropism? epitheliotropism? Was there IAV NP positivity seen**
**in nuclei and/or cytoplasm of infected cells? In order to find answers or give some indications on the**
**route of infection and/or pathogenesis? Please elaborate on this.**

Autolytic changes complicated proper and detailed assessment of cellular and subcellular distribution
for both viral antigen and RNA. However, we agree with the reviewer about the importance of
providing an overview of viral tissue distribution, in regards of pathogenesis/route of infection. Upon
reevaluation of histopathological slides, we were able to provide the subtissular localization of viral
antigen/RNA for each organ examined, as detailed in Supplementary tables 2 and 3. Based on these
findings, in terms of pathogenesis we could conclude that:

- 1. Endotheliotropism is minimal
2. Neurotropism is supported by positivity of peripheral visceral ganglia. This finding is consistent with
viral antigen/RNA tissue distribution identified in HPAI naturally and experimentally-infected avian
species, such as bustards and domestic ducks, object of previous studies.
3. Epitheliotropism varies according to the organ, but proper assessment is limited by autolytic
changes.
4. Overall, we identified some common pathogenic traits with domestic avian species at a similar stage
of infection.

More specific comments and questions (some may overlap with general questions):

-L19: consider for consistency with TBB (*Ursus thibetanus*), also to include Latin name of BHGU (*Chroicocephalus ridibundus*) already here in abstract?

The manuscript was modified accordingly.

L22: closest related one.. unclear ending, to what/which?

The ending is now: *"the closest related strain"*.

L42: The high (and low) pathogenicity phenotype is defined in chickens (poultry).

The manuscript was modified accordingly.

L47: Gallinacea, consider to include also generic layman names here such as 'chickens, poultry, and/or land fowl' or similar? In order to clarify to the reader that maybe not so familiar with technical taxonomical nomenclature.

Please see line 70 of this letter.

L78-80: any clinical nervous signs observed? If so what kind, severity, duration? Please provide more detail.

Please, see lines 183-185 of this letter.

L80: What kind of analysis, blood? Antemortem (how long before death?) or postmortem sampled?

The tests involved blood samples taken the day before the bear's death. The manuscript was modified accordingly.

L80: is there an explanation for the decubitus found? Was it an acute or chronic lesion? Due to abnormal (nervous) scratching/rubbing behavior? Were the other TBBs not affected by decubitus? Was it a specific lesion?

The bear developed clinical signs the day before it died. The vets noted afterwards that he had been less energetic for a few days, but this did not worry them. On clinical examination the day before death, the bear was in severe dyspnoea, very depressed and hypertermic (rectal temperature was 39°C), which may well explain the decubitus.

L84: what kind of gulls? BHGUs?

It was only one gull (the one that were necropsied). Please see Supplementary Table 1.

**L87: space**

The manuscript was modified accordingly.

**L88: other TBBs, were they housed in the same enclosure? Also dead birds found with other bears?**
**Interspecies transmission between TBBs suspected/likely/unlikely/impossible?**

This paragraph has been modified and the addition of supplementary table 1 makes things clearer. We
can only speculate about inter-bear transmission, since the other bears, to our regret, were not
sampled for testing. This is now noted in the discussion section (lines 181-183).

**L82-84: Gross lesions, can you be more specific, stomach contents? Bird remains in the stomach? Or**
**in other segments of the GI tract? Which lymph node is meant with visceral ln? From thorax or**
**abdomen? Which particular ln was it? Draining form lungs/respiratory tract, upper or lower, or**
**draining from liver or GI tract?**

Please, see lines 164-167 of this letter. Unfortunately, the exact location of the lymph node was not
noted during autopsy. The specimen label only stated "visceral lymph node".

**L105: Which cells were positive by IHC in lung interstitium/parenchyma? Endothelium or epithelium**
**or others? Idem glomerular tuft, what cells? Idem myocardium? Myocardiocytes or endothelium or**
**other cells? Was the vasculitis associated with NP positive endothelial cells by IHC? These are**
**important facts of information to interpret and possibly (partly) unravel the pathogenesis.**

The subtissular localization for lungs, kidney and myocardium is listed in supplementary tables 2 and
3. Viral antigen detection was associated with vasculitis but it was not localized to the endothelial cells,
rather to the leucocytes and perivascular tissue adjacent to the necrotic foci. Viral antigen and RNA
detection was rarely and sparsely observed (supplementary table 2)

**L112: Myocardial atherosclerosis and mineralisations? What extent? And location, which blood**
**vessels/compartments, severe enough to cause disease/death? Were these diagnosed as co-**
**morbidities of importance?**

Please, see lines 131-144 of this letter.

**L114-117: what cells were infected or positive for NP by IHC and/or positive by ISH? Morphologically**
**consistent with which cells? Were the inflamed organs with intralesional presences of viral**
**protein/RNA interpreted as the aetiology of morbidity/mortality? Ok some listed in supplement but**
**why not include this important info in results section?**

Please, see supplementary table 3. These findings were also included in the manuscript.

**L166: Indeed, the important question is whether the TBB had contact with a dead bird as possible**
**source of infection, so like previously is there evidence from the BHGU carcass in the enclosure or in**
**the bear GI tract/feces to support this speculation or not?**

Please, see lines 164-167 of this letter.

**L171-173: which lymph node? So it drained from the respiratory tract?**

Please, see lines 267-268 of this letter.

**L192: animal species**

The manuscript was modified accordingly.

**Supplementary figures 1-3: Tissues from a Tibetan black bear, not blue bear, also in legend. And**
**official nomenclature is black-headed gull (BHGU), not seagull, also in legend.**

Thank you for spotting these mistakes. The manuscript was modified accordingly.

Re: Spectrum03736-23R1 (High pathogenicity avian influenza A (H5N1) clade 2.3.4.4b virus infection in a captive Tibetan black bear (*Ursus thibetanus*): investigations based on paraffin-embedded tissues, France, 2022)

Dear Dr. Pierre Bessière:

Your manuscript has been accepted, and I am forwarding it to the ASM production staff for publication. Your paper will first be checked to make sure all elements meet the technical requirements. ASM staff will contact you if anything needs to be revised before copyediting and production can begin. Otherwise, you will be notified when your proofs are ready to be viewed.

Sincerely,
Robert de Vries
Editor
Microbiology Spectrum